# Is It Possible to Notice the Unmet Non-Medical Needs among Cancer Patients? Application of the Needs Evaluation Questionnaire in Men with Lung Cancer

Karolina Osowiecka [1,*], Marcin Kurowicki [2], Jarosław Kołb-Sielecki [3], Anna Gwara [4], Marek Szwiec [5], Sergiusz Nawrocki [6] and Monika Rucińska [6]

1. Department of Psychology and Sociology of Health and Public Health, School of Public Health, University of Warmia and Mazury in Olsztyn, Warszawska 30, 10-082 Olsztyn, Poland
2. Department of Radiotherapy, NU-MED Radiotherapy Center in Elblag, Królewiecka 146, 82-300 Elblag, Poland
3. Department of Oncology, The Center for Pulmonary Diseases in Olsztyn, Jagiellońska 78, 10-357 Olsztyn, Poland
4. Department of Nursing, Faculty of Medicine and Health Sciences, University of Zielona Gora, ul. Zyty 28, 65-046 Zielona Gora, Poland
5. Department of Surgery and Oncology, Faculty of Medicine and Health Sciences, University of Zielona Gora, Zyty 28, 65-046 Zielona Gora, Poland
6. Department of Oncology, Collegium Medicum University of Warmia and Mazury in Olsztyn, Wojska Polskiego 37, 10-228 Olsztyn, Poland
* Correspondence: karolina.osowiecka@uwm.edu.pl; Tel.: +48-518-711-334

**Abstract:** Background: Lung cancer is the most common cause of cancer death worldwide. It is the most frequently diagnosed cancer in men. Lung cancer causes not only physical symptoms related to the disease itself and its treatment but also numerous mental, social and spiritual problems. The aim of the study was to assess non-medical needs among male lung cancer patients during oncological treatment. Materials and Methods: The study was conducted on a group of 160 men (mean age 67 years) treated for lung cancer from June 2022 until November 2022 in 5 oncological centers in Poland. The Needs Evaluation Questionnaire (NEQ) was used. The NEQ explores five areas of patients' needs: informative, connected with assistance/care, relational, material and psycho-emotional support. Results: All participants (except one) expressed some unmet non-medical needs (mean and median 11). Male lung cancer patients indicated informative needs most frequently. There were no significant differences between expressed unmet needs based on age, place of residence, professional activity or marital status. Conclusions: The NEQ seems to be a proper instrument to explore the non-medical needs of cancer patients. Adequate measures to address the unmet needs of lung cancer patients could contribute to an improved quality of life.

**Keywords:** men's health; lung cancer; non-medical needs; NEQ; quality of life

## 1. Introduction

Cancer is a worldwide civilization disease. According to WHO [1], there were 19.3 million newly diagnosed cancer cases and nearly 10 million deaths caused by cancer reported in 2020. Cancer is the second most prevalent cause of death in the world as a whole and also in Poland [1,2]. In some countries (e.g., Japan, Denmark), cancer mortality has become the leading cause of death [2–4]. Lung cancer was the second most common cancer (over 2 million cases per year) and the most common cause of cancer death (approximately 1.80 million deaths per year) worldwide in 2020 [1]. In Poland, lung cancer ranks first among the causes of cancer deaths in both men and women. However, lung cancer is more common in men. In Poland, among men, 16.1% of new cancer cases were lung cancer compared to 9.9% in women. Lung cancer accounted for 27.4% of cancer deaths in men

and 17.9% in women in 2019 [5]. The survival prognosis for patients with lung cancer is poor. According to EUROCARE-5 [6], the 5-year relative survival rates were only 14.3% and 12.6%, respectively, for Poland and Europe.

Lung cancer causes numerous physical symptoms, both related to the disease itself and its treatment. Lung cancer patients may also experience mental, social and spiritual problems. People faced with a traumatic event, such as a diagnosis of lung cancer, often need psychological support—they may look for it both within themselves and from those around them. Patients use different cognitive strategies for psychological self-support. Cognitive processes can be helpful in regulating emotions after stressful experiences, for example, cognitive strategies including acceptance, positive refocusing, refocusing on planning or positive reappraisal [7]. However, lifestyle choices associated with an increased risk of lung cancer, may cause patients to develop destructive cognitive strategies such as self-blame, rumination, catastrophizing or blaming others. The main risk factor for developing lung cancer is smoking, which is responsible for about 80% of lung cancer diagnoses. People who smoke also often abuse alcohol [5]. Our previous study showed that lung cancer patients significantly most frequently used catastrophizing and rumination during oncological treatment [8]. Weiss et al. [9] demonstrated that smoking was a significant predictor of personal stigmatization among patients with lung cancer. Additionally, the multivariable model showed that one predictor of self-blame in patients was the belief that smoking was a cause of their lung cancer [9].

Help from specialists (e.g., psychologists, psycho-oncologists, social workers or spiritual advisors) may be needed to modify the cognitive processes of patients for the improvement of their mental health. During treatment, cancer patients should have access not only to clinicians but also to psychologists, spiritual advisors and social workers. Each cancer patient is also surrounded by individuals who may offer informal support, e.g., immediate family, other relatives, friends and other patients. However, there is a doubt that patients are receiving the appropriate support they need. Patients can get the support they need if they express their needs and if their needs are noticed by others. Sometimes it is difficult for patients to identify and express their non-medical needs. Patient organizations can be helpful in providing support. In Poland, there are different patients' organizations such as "Amazons", which connects women with breast cancer, and "Gladiator" for men with prostate cancer, but there are no patients' organizations to support and connect lung cancer patients.

Patients with a diagnosis of lung cancer encounter numerous problems during oncological treatment. Symptoms of the disease and side effects of therapies may cause suffering, pain and disability. Patients encounter and have to overcome new life experiences, such as an unfamiliar oncology system, a lack of understanding of their condition and the proposed treatment, and uncertainty about the future. Patients can feel lost and need information, as well as psychological and material support. Hsieh et al. [10] demonstrated that the need for some information among lung cancer patients was the highest at the time of diagnosis. The need for information about the disease itself was the most commonly expressed by patients [10,11]. Moreover, at the time of cancer diagnosis patients feel helplessness, anxiety and depression [12]. It is unclear how familiar medical staff are with patient expectations. Awareness of patients' non-medical needs could help to bridge the gap between patient expectations and the services that a healthcare system provides.

The aim of this study was to assess the non-medical needs of male lung cancer patients during oncological therapy, including informative needs, psycho-emotional needs, relational needs, material needs and needs related to assistance/care.

## 2. Materials and Methods

### 2.1. Participants

The study was carried out on a group of 160 men treated for lung cancer from June 2022 until November 2022 in five oncological centers in Poland (University Hospital in Zielona Gora, Hospital of the Ministry of Internal Affairs with Warmia and Mazury Oncology

Center in Olsztyn, The Center for Pulmonary Diseases in Olsztyn, Hospital in Prabuty, NU-MED Radiotherapy Center in Elbląg).

The inclusion criteria were men, aged ≥18 years old, pathologically confirmed lung cancer diagnosis, oncology treatment ongoing or having finished treatment no longer than 3 months previously, current hospitalization for at least 3 days or after at least one hospitalization due to oncological treatment within the previous 3 months. The criterion of hospitalization was chosen so that patients have experienced possible problems related to hospitalization. Exclusion criteria were performance status >2 according to the Eastern Cooperative Oncology Group (ECOG) scale and the patient's psychophysical inability to complete the questionnaire.

Questionnaire

The Needs Evaluation Questionnaire (NEQ) was originally designed and validated by Tamburini et al. [13,14] at the Psychology Unit of Instituto Nazionale Tumori in Italy. In this research, the Polish version of the NEQ was used to assess the unmet needs of male lung cancer patients. A back-translation procedure was used to develop the Polish version of NEQ. Two independent professional translators translated the English version into Polish, and then the Polish version was translated back into English by two other translators. The final version of the NEQ was discussed with the translators and reviewed with a psycho-oncologist and a public health expert. The NEQ was previously validated among the Polish population of cancer patients [15]. The Polish version of the NEQ showed good reliability, acceptability, comprehensibility and structure validity [15]. Polish and English versions of the questionnaire are presented as Supplementary Materials.

The study protocol was approved by the Ethics Committee of the University of Warmia and Mazury in Olsztyn (No. 30/2020). Participation in the study was voluntary. All study participants were informed about the aim of the study and gave their consent and signed it.

## 2.2. Statistical Analysis

The percentages of various needs (23 items of NEQ) were calculated. Five subgroups of needs (informative needs, needs related to assistance/care, relational needs, needs for psycho-emotional support and material needs) were distinguished according to Annunziata et al. [16]. Medians with interquartile ranges of the level of needs in these subgroups were assessed. The normal distribution of continuous variables was tested using the Shapiro–Wilk test. The differences in prevalence of each need, assessed by the NEQ, based on demographic factors (age, education, place of residence, professional activity, marital status, living with, doctor as close family or friend) were determined using the t-student test and the chi-square test for continuous and categorical variables respectively. The correlations between five subgroups of needs and demographic factors were assessed using Spearman's rank correlation coefficient, Mann–Whitney test (for 2 subgroups) or Kruskal–Wallis test (for >2 subgroups). A value of $p < 0.05$ was considered to be significant. The data analysis was conducted using R studio version 4.0.2 (22 June 2020).

## 3. Results

### 3.1. Characteristics of Patients

The study was conducted on a group of 160 men with ages between 44 and 88 years old (mean age 66.9 years). All participants were diagnosed with lung cancer. Most patients had graduated from secondary school (66.9%), lived in cities (62.5%), were pensioners (73.1%), married (76.3%) and living with a partner (70.6%). Only 12.5% of respondents had a doctor in their close family or among friends (Table 1).

**Table 1.** Study group demographics.

|  |  | *n* | % |
|---|---|---|---|
| Age range 44–88 years, mean ± SD | 66.9 ± 8.0 |  |  |
| Education |  |  |  |
|  | primary | 38 | 23.7 |
|  | secondary | 107 | 66.9 |
|  | high | 14 | 8.8 |
|  | no data | 1 | 0.6 |
| Place of residence |  |  |  |
|  | urban | 100 | 62.5 |
|  | rural | 60 | 37.5 |
| Professional activity |  |  |  |
|  | active | 37 | 23.1 |
|  | unemployed | 6 | 3.8 |
|  | pensioner | 117 | 73.1 |
| Marital status |  |  |  |
|  | married | 122 | 76.3 |
|  | relationship broken down during disease or in relation to disease | 1 | 0.6 |
|  | single | 36 | 22.5 |
|  | no data | 1 | 0.6 |
| Living with |  |  |  |
|  | partner | 113 | 70.6 |
|  | child/children/another family member | 20 | 12.5 |
|  | alone | 21 | 13.1 |
|  | no data | 6 | 3.8 |
| Doctor in close family or friends |  |  |  |
|  | yes | 20 | 12.5 |
|  | no | 138 | 86.2 |
|  | no data | 2 | 1.3 |

±SD—standard deviation.

### 3.2. Needs Prevalence

The NEQ explores five areas of patients' needs: informative, connected with assistance/care, relational, material and psycho-emotional support needs. All participants (except one) expressed one or more unmet non-medical needs (mean 10.6; median 11); 74% of lung cancer patients answered Yes to more than five needs assessed in the NEQ. Six patients (3.8%) claimed that all 23 of the needs being assessed were not met. Male lung cancer patients indicated unmet informative needs most frequently (Table 2).

**Table 2.** Male lung cancer patients' needs in five areas of unmet needs.

| Area of Unmet Need | Median (25–75% IQR) |
|---|---|
| Informative needs | 0.7 (0.2–0.9) |
| Needs related to assistance/care | 0.0 (0.0–0.3) |
| Material needs | 0.3 (0.3–0.7) |
| Relational needs | 0.5 (0.0–0.8) |
| Needs for psycho-emotional support | 0.3 (0.0–0.3) |

IQR—interquartile range.

More than half of respondents expressed the need for more information about their diagnosis (55.0%), about the examinations they are undergoing (55.0%), about their treatments (59.4%) and about their future condition (67.5%). Half of patients expressed the need to be more involved in therapeutic choices (50.6%). More than half of male lung cancer patients in the study expressed communication-related needs to have a better dialogue with clinicians (52.5%), for clinicians and nurses to give more comprehensible information

(58.7%), for clinicians to be more sincere (61.3%) and to be more reassured by clinicians (58.7%). With regard to the needs related to assistance/care, 58.7% of patients needed their symptoms (pain, nausea, insomnia, etc.) to be better controlled. One-third of patients (31.2%) indicated that they need better attention from nurses. A quarter of patients (25.6%) needed better respect for their intimacy. Although 15.6% of respondents needed more help with eating, dressing and going to the bathroom, 59.4% of the men in the study expressed the need for better services from the hospital (bathrooms, meals, cleaning); 60.0% of patients needed to have more information about economic insurance in relation to their illness. A third of patients (29.4%) indicated the need for economic help. In the category of needs relating to psycho-emotional support, 21.3% of lung cancer patients expressed the need to speak with a psychologist, 26.3% of respondents indicated the need to speak with a spiritual advisor and 41.9% of them needed to speak with people who have had a similar experience. Male lung cancer patients also expressed relational needs. More than half of respondents (57.5%) needed to feel more useful within their families. Some patients needed to be more reassured by their relatives (43.1%), and some needed to receive less commiseration from other people (40.6%). One-third of respondents expressed the need to feel less abandoned (33.1%) (Table 3).

**Table 3.** The distribution of patients' needs.

| | Item Number | | *n* | % |
|---|---|---|---|---|
| Q1 | I need more information about my diagnosis | yes | 88 | 55.0 |
| | | no | 72 | 45.0 |
| Q2 | I need more information about my future condition | yes | 108 | 67.5 |
| | | no | 51 | 31.9 |
| | | missing data | 1 | 0.6 |
| Q3 | I need more information about the exams I am undergoing | yes | 88 | 55.0 |
| | | no | 67 | 41.9 |
| | | missing data | 5 | 3.1 |
| Q4 | I need more explanations of treatments | yes | 95 | 59.4 |
| | | no | 60 | 37.5 |
| | | missing data | 5 | 3.1 |
| Q5 | I need to be more involved in the therapeutic choices | yes | 81 | 50.6 |
| | | no | 72 | 45.0 |
| | | missing data | 7 | 4.4 |
| Q6 | I need clinicians and nurses to give me more comprehensible information | yes | 94 | 58.7 |
| | | no | 63 | 39.4 |
| | | missing data | 3 | 1.9 |
| Q7 | I need clinicians to be more sincere with me | yes | 98 | 61.3 |
| | | no | 57 | 35.6 |
| | | missing data | 5 | 3.1 |
| Q8 | I need to have a better dialogue with clinicians | yes | 84 | 52.5 |
| | | no | 71 | 44.4 |
| | | missing data | 5 | 3.1 |
| Q9 | I need my symptoms (pain, nausea, insomnia, etc.) to be better controlled | yes | 94 | 58.7 |
| | | no | 62 | 38.8 |
| | | missing data | 4 | 2.5 |
| Q10 | I need more help with eating, dressing and going to the bathroom | yes | 25 | 15.6 |
| | | no | 131 | 81.9 |
| | | missing data | 4 | 2.5 |

**Table 3.** *Cont.*

| Item Number | | | *n* | % |
|---|---|---|---|---|
| Q11 | I need better respect for my intimacy | yes | 41 | 25.6 |
| | | no | 114 | 71.3 |
| | | missing data | 5 | 3.1 |
| Q12 | I need better attention from nurses | yes | 50 | 31.2 |
| | | no | 104 | 65.0 |
| | | missing data | 6 | 3.8 |
| Q13 | I need to be more reassured by the clinicians | yes | 94 | 58.7 |
| | | no | 59 | 36.9 |
| | | missing data | 7 | 4.4 |
| Q14 | I need better services from the hospital (bathrooms, meals, cleaning) | yes | 95 | 59.4 |
| | | no | 58 | 36.2 |
| | | missing data | 7 | 4.4 |
| Q15 | I need to have more economic insurance information (tickets, invalidity, etc.) in relation to my illness | yes | 96 | 60.0 |
| | | no | 60 | 37.5 |
| | | missing data | 4 | 2.5 |
| Q16 | I need economic help | yes | 47 | 29.4 |
| | | no | 108 | 67.5 |
| | | missing data | 5 | 3.1 |
| Q17 | I need to speak with a psychologist | yes | 34 | 21.3 |
| | | no | 121 | 75.6 |
| | | missing data | 5 | 3.1 |
| Q18 | I need to speak with a spiritual advisor | yes | 42 | 26.3 |
| | | no | 111 | 69.3 |
| | | missing data | 7 | 4.4 |
| Q19 | I need to speak with people who have this same experience | yes | 67 | 41.9 |
| | | no | 88 | 55.0 |
| | | missing data | 5 | 3.1 |
| Q20 | I need to be more reassured by my relatives | yes | 69 | 43.1 |
| | | no | 85 | 53.1 |
| | | missing data | 6 | 3.8 |
| Q21 | I need to feel more useful within my family | yes | 92 | 57.5 |
| | | no | 64 | 40.0 |
| | | missing data | 4 | 2.5 |
| Q22 | I need to feel less abandoned | yes | 53 | 33.1 |
| | | no | 99 | 61.9 |
| | | missing data | 8 | 5.0 |
| Q23 | I need to receive less commiseration from other people | yes | 65 | 40.6 |
| | | no | 90 | 56.3 |
| | | missing data | 5 | 3.1 |

*3.3. Correlations*

The needs expressed by male lung cancer patients who participated in this study generally were not significantly correlated with specific demographic factors. There were no significant differences between expressed needs (23 items of the NEQ) based on age, place of residence, professional activity or marital status. Patients who had graduated from high school most often expressed the need for more help with eating, dressing and going to the bathroom compared with patients who were less highly educated ($p = 0.02$) (35.7%, 23.7% and 10.7%, respectively, for high, primary and secondary schools). Men who were living alone indicated the need for more help with eating, dressing and going to the bathroom (35%) than those who were living with a partner (12.6%) and with a child/children or another family member (10.5%) ($p = 0.03$). The need to have more

information about economic insurance information in relation to their illness was the most frequently expressed in patients who were living with a child/children or another family member (73.7%) (*p* = 0.028). Respondents who did not have a doctor in their close family or among friends more frequently expressed the need for additional explanation of treatments than those who knew a doctor, 64.2% and 36.8%, respectively (*p* = 0.04) (Table S1 presented in Supplementary Materials).

## 4. Discussion

Currently, a holistic and personalized approach to the treatment and care of cancer patients is recommended. Patient well-being has become an important outcome of cancer treatment. Emotional, social and spiritual well-being is important for improving quality of life [17]. Cancer is associated with distress, which is recognized as a factor worsening quality of life. Meeting patients' non-medical needs can decrease distress and reduce their suffering [18]. Unmet needs are related to hope—meeting unmet needs encourages hope in cancer patients [19]; therefore, it is important to notice and to pay attention to patient expectations. There is a lack of studies concerning the non-medical needs of patients, especially among lung cancer patients.

In the current study, the prevalence of male lung cancer patients' needs was determined using the Needs Evaluation Questionnaire (NEQ). The NEQ was previously used to evaluate the needs of patients with diverse cancer diagnoses [14,20,21]. The relationship between unmet needs and the level of distress and hope among patients was investigated [14,19–21]. Bonacchi et al. [21] demonstrated a significant correlation between unmet needs among cancer patients and their level of distress. Unmet informative needs, needs related to assistance, material needs, relational needs and needs for psycho-emotional support were significantly associated with lower resilience [21]. However, there is a lack of studies concerning the non-medical needs of lung cancer patients. Sanders et al. [22] found that lung cancer patients with higher supportive care needs expressed a higher level of physical distress and intrusive thoughts about cancer. These patients also experienced greater symptom bother and a lower level of satisfaction with health care [22]. Rapamonti et al. [19] determined significant correlations between higher levels of unmet needs in patients and lower levels of hope (with respect to having sincere clinicians, better dialogue with and being more reassured by, clinicians, respect for intimacy, attention from nurses, speaking with people who have had a similar experience, being more reassured by relatives, feeling less abandoned).

The current study showed that male lung cancer patients reported a range of non-medical needs; however, informative needs were most often expressed. The average proportion of unmet non-medical needs from those assessed by the NEQ was almost 50% (11 of the 23 needs assessed), and 74% of study participants expressed more than five unmet needs. Tamburini et al. [14] postulated that expressing more than five needs in the NEQ indicates that a cancer patient demonstrates great skill in recognizing and expressing experiences relative to their condition. Male lung cancer patients in this study expressed informative needs most frequently. Studies of patients with diverse cancer types showed that a significantly higher degree of unmet needs were reported by lung cancer patients than those with other types of cancer [23,24]. Lung cancer patients were more likely to emphasize unmet psychological needs and daily living needs compared to patients with other types of cancer [23]. Needs of an informational nature are frequently reported by lung cancer patients [23]. Lewandowska et al. [12], using an alternative questionnaire, demonstrated that, in general, Polish cancer patients expressed predominantly informative needs. Among Italian cancer patients, the informative needs reported using the NEQ were also the most frequently underlined unmet non-medical needs [14,20]. This trend was also observed among lung cancer patients [14,20,21].

More than half of male lung cancer patients in the current analysis expressed the need for more information about their diagnosis, examinations they are undergoing, treatments and their prognosis. The need for more information about their future conditions was expressed by 50–60% of cancer patients [14,20,22]. Mistry et. al. [25], using a different

questionnaire showed that up to 90% of cancer patients had a high degree of information needs regarding their prognosis. In the current study, 67.5% of male lung cancer patients needed more information about their future conditions. Hsieh et al. [10] noted that lung cancer patients diagnosed with stage III disease expressed a significantly higher level of informative needs than patients with cancer at stages I–II. In general, patients have considerable informative needs at the time of diagnosis, but these decrease during treatment [10,25,26]. However, the need for information significantly increased over time among patients diagnosed with stage IV lung cancer [10]. This could be related to anxiety about their prognosis. Younger patients reported a greater need for information regarding their future [11,27]. Hsieh et al. [10] found that lung cancer patients with a higher level of education needed more psychosocial-related information than patients who were less well educated [10]. However, other authors did not confirm the association between increased supportive care needs and information needs of lung cancer patients and level of education or living with someone/alone [11]. In the current study, the association between informative needs and level of education was not observed.

In the present study, 64% of respondents who did not have a doctor in their close family or among friends needed more explanation of treatments than those who knew a doctor. Some lung cancer patients would like to have someone from the hospital staff with whom they can talk about all aspects of their condition [11].

In this study, half of male lung cancer patients expressed the need to be more involved in therapeutic choices and to have a better dialogue with clinicians, whereas in studies among Italian cancer patients, these needs were expressed by about 40% of respondents [14,20]. More than 80% of lung cancer patients from the Netherlands found it important to be involved in decisions relating to their treatment [28]. High-quality patient–clinician communication and/or better patient information and/or involvement in treatment decision making could be associated with less decisional conflict [28,29].

In this study, 58.7% of patients needed their symptoms (pain, nausea, insomnia, etc.) to be better controlled. Lung cancer patients often suffer from problems such as shortness of breath, cough, lack of energy, fatigue and tiredness. Some authors reported that the informative need relating to managing these symptoms is important for 63% of lung cancer patients [11,22].

The need for more support with daily activities was not prevalent—only 15.6% of male lung cancer patients in the current study expressed the need to have more help with eating, dressing and going to the bathroom. Among Italian cancer patients, the results were similar—about 10% of respondents needed this support [14,20]. In the present study, patients who were living alone indicated a greater need for help with daily activities than patients living with someone. More educated respondents significantly more frequently expressed the need for support in general functioning when compared with less educated patients. However, Bonacchi et al. [21] reported that, among Italian cancer patients, the lower level of education was correlated with increased needs related to assistance.

In the current study, 60% of male lung cancer patients expressed the need for better services from the hospital and more information about economic insurance in relation to their illness. Italian authors reported these needs among about 40% of cancer patients [14,20]. In the present study, patients who were living with a child/children or another family member more commonly needed economic insurance information in relation to their illness than patients living with a partner or alone. Bonacchi et al. [21] reported that younger cancer patients had significantly more material support needs in comparison with older patients. Younger patients had a greater demand for information on insurance options and possible future employment than older patients [11,27]. This correlation was not found in the current study.

In this study, 21% of male lung cancer patients expressed the need to speak with a psychologist, and 26% of respondents indicated the need to speak with a spiritual advisor. Similar results were shown among Italian cancer patients, but there was a lower level of need for support from a priest (expressed by 15% of respondents) [14,20]. In another

Polish study, 5% of cancer patients indicated the need to receive spiritual support from a priest [12]. Giuliani et al. [11] reported that 66% of lung cancer patients in Canada needed more psychological support. Li and Girgis [23] observed higher levels of unmet psychological needs among lung cancer patients compared with other cancer patients. However, Bonacchi et al. [21] reported that breast cancer patients significantly more often required psycho-emotional support than patients with lung, colon or stomach cancer. Younger and single patients needed significantly more psycho-emotional support than older and married patients [21]. There were no differences between the need to speak with a psychologist or a spiritual advisor based on the age or marital status of male lung cancer patients in the current study. Moreover, 42% of male lung cancer patients expressed the need to speak with people who have had a similar experience. However, only 20% of all Polish cancer patients in the study conducted by Lewandowska et al. [12] would like to join a support group.

## 5. Conclusions

The NEQ seems to be a proper instrument to explore the non-medical needs of cancer patients. Unmet non-medical needs are common in men with lung cancer. Almost all participants expressed one or more unmet non-medical needs. Male lung cancer patients indicated unmet informative needs most frequently. Meeting patients' needs could result in physical and psychological well-being and even increased hope. Adequate measures to address the non-medical needs of lung cancer patients will contribute to an improved quality of life.

**Supplementary Materials:** The following supporting information can be downloaded at: https://www.mdpi.com/article/10.3390/curroncol30030264/s1, Table S1; NEQ—English version; NEQ—Polish version.

**Author Contributions:** Conceptualization, K.O. and M.R.; methodology, K.O. and M.R.; validation, K.O. and M.R.; formal analysis, K.O. and M.R.; investigation, K.O., M.K., J.K.-S., A.G., M.S., S.N. and M.R.; data curation, K.O. and M.R.; writing—original draft preparation, K.O. and M.R.; writing—review and editing, M.K., J.K.-S., A.G., M.S. and S.N.; visualization, K.O. and M.R. All authors have read and agreed to the published version of the manuscript.

**Funding:** The research was funded in part by the National Science Centre, Poland. Grant number 2022/06/X/NZ7/00803.

**Institutional Review Board Statement:** The study was conducted in accordance with the Declaration of Helsinki, and approved by the Ethics Committee of the University of Warmia and Mazury in Olsztyn (No. 30/2020; 1 June 2020).

**Informed Consent Statement:** Informed consent was obtained from all subjects involved in the study. Written informed consent has been obtained from the patient(s) to publish this paper (a blank copy of the consent form is included in Supplementary Materials).

**Data Availability Statement:** The datasets used and/or analyzed during the current study are available from the corresponding author upon reasonable request.

**Conflicts of Interest:** The authors declare no conflict of interest.

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
