# Peer review of "Is It Possible to Notice the Unmet Non-Medical Needs among Cancer Patients? Application of the Needs Evaluation Questionnaire in Men with Lung Cancer"

_curroncol, doi:10.3390/curroncol30030264_

Round 1

Reviewer 1 Report

First of all, I'd like to ask the authors why do you chose to interview only male patients with lung cancer.
Even if there are more male patients with lung cancer,  the amount of women diagnosed with lung cancer is incresasing worldwilde.
Also, women rarely have a full support at home , regarding their disease, since they are usually the caretaker of other family members, so they probably have more unmet needs that they need to be addressed. I think the authors should take that in to account.

Also, what do you mean about "spiritual need"? It seems unclear to me in the paper.

Author Response

Thank you very much for your comments.

We would like to explain as follows:

Current study is an initial study. Therefore, after the validation of NEQ, we decided to use this questionnaire in a homogeneous group. The study was conducted in lung cancer patients, because lung cancer is the most frequent cancer worldwide. We chose male lung cancer patients due to belief that men could be patients with more unmet needs – men probably have some difficulties in expressing their needs, are more embarrassment with asking for support, would like to be brave... We are planning an extending study by inclusion patients with other cancer types. Of course women also will be included into future study. The gender will be one of the factor analyzed in multivariate model.

In the study we avoided to use phrase “spiritual need”. We divided unmet needs into five areas according to Annunziata et al. “Spiritual needs” are included in “needs for psycho-emotional support”. In the study we only discussed the need to speak with a “spiritual advisor”, for example priest. “Spiritual support” could be related to talk about sense of life, God, idea, end of life, ect.

Reviewer 2 Report

The authors have done a questionnaire-based research on the unmet non-medical needs among cancer patients diagnosed with lung cancers. The study performed is very interesting and very well documented.

My recommendations are as follows:

ABSTRACT

The abstract is well written. Please Delete the (1), (2),…etc in front of each chapter title.

Please correct with materials and methods for that part (instead of only methods).

The conclusion of the study should be an answer to your title which serves as the hypothesis of the study.

INTRODUCTION:

The introduction is well written and well documented through bibliographic references.

No further comments on this part.

MATERIALS AND METHODS:

Please specify if there is also an exclusion criteria (line 102)

Please specify if the polish version of the NEQ questionnaire is exactly the same as the English version, except the language.

RESULTS:

The results part is very well written.

On the first row of each column, please change the n-160 to only n because on that row you define the unit of measure (either in numbers, either in percentage).

I suggest to change the city and village words to urban and rural.

DISCUSSIONS:

The part of discussions is well written and very well explained.

CONCLUSIONS:

Please be more specific in this part. Your conclusion is, in my opinion, too general.

REFERENCES:

Over half of the reference are older than five years. Please try to find a little more recent references.  

Author Response

Thank you very much for your comments.

We would like to explain as follows:

ABSTRACT

The abstract is well written. Please Delete the (1), (2),…etc in front of each chapter title.

  • We deleted, because there was misunderstanding in the instructors for authors.

Please correct with materials and methods for that part (instead of only methods).

  • We corrected.

The conclusion of the study should be an answer to your title which serves as the hypothesis of the study.

  • We corrected.

MATERIALS AND METHODS:

Please specify if there is also an exclusion criteria (line 102)

  • We added.

Please specify if the polish version of the NEQ questionnaire is exactly the same as the English version, except the language.

  • Yes, a professional, appropriate translation has been done. We added explanation.

RESULTS:

The results part is very well written.

On the first row of each column, please change the n-160 to only n because on that row you define the unit of measure (either in numbers, either in percentage).

  • We corrected.

I suggest to change the city and village words to urban and rural.

  • We corrected.

CONCLUSIONS:

Please be more specific in this part. Your conclusion is, in my opinion, too general.

  • We corrected.

REFERENCES:

Over half of the reference are older than five years. Please try to find a little more recent references.  

  • We conducted a systematic search in the PUBMED database. There is a lack of studies related to this subject.

Round 2

Reviewer 2 Report

Dear authors. 

All my concerns were answered. 

Thank you.